# Study of the Strengths and Weaknesses of Nursing Work Environments in Primary Care in Spain

**DOI:** 10.3390/ijerph18020434

**Published:** 2021-01-07

**Authors:** Vicente Gea-Caballero, José Ramón Martínez-Riera, Pedro García-Martínez, Jorge Casaña-Mohedo, Isabel Antón-Solanas, María Virtudes Verdeguer-Gómez, Iván Santolaya-Arnedo, Raúl Juárez-Vela

**Affiliations:** 1Nursing School La Fe, Adscript Center of Universidad de Valencia, 46026 Valencia, Spain; gea_vic@gva.es; 2Research Group GREIACC, Health Research Institute La Fe, Avda. Fernando Abril Martorell, 106, Pabellón Docente Torre H, Hospital La Fe, 46026 Valencia, Spain; 3Departamento Enfermería Comunitaria, Medicina Preventiva y Salud Pública e Historia de la Ciencia, Universidad de Alicante, 03080 Alicante, Spain; 4Health Department, Universidad Católica de Valencia, C/Quevedo 2, 46001 Valencia, Spain; jorge.casana@ucv.es; 5Department of Physiatry and Nursing, Faculty of Health Sciences, University of Zaragoza, C/Domingo Miral s/n, 50009 Zaragoza, Spain; ianton@unizar.es; 6Research Group Nursing Research in Primary Care in Aragón (GENIAPA) (GIIS094), Institute of Research of Aragón, Avenida San Juan Bosco, 13, 50009 Zaragoza, Spain; 7Departament de Salut La Ribera, Atención Primaria, Ctra. Corbera, Alzira, 46600 Valencia, Spain; mariviverdeguer@gmail.com; 8Centro de Investigación Biomédica de la Rioja, Logrono, 26006 La Rioja, Spain; isantolalla@riojasalud.es (I.S.-A.); raul.juarez@unirioja.es (R.J.-V.); 9Department of Nursing, University of La Rioja, Logroño, 26006 La Rioja, Spain

**Keywords:** nursing, primary care, workplace, quality of health care, nurse’s role

## Abstract

Background: Nursing work environments are defined as the characteristics of the workplace that promote or hinder the provision of professional care by nurses. Positive work environments lead to better health outcomes. Our study aims to identify the strengths and weaknesses of primary health care settings in Spain. Methods: Cross-sectional study carried out from 2018 to 2019. We used the Practice Environment Scale of the Nursing Work Index and the TOP10 Questionnaire of Assessment of Environments in Primary Health Care for data collection. The associations between sociodemographic and professional variables were analyzed. Results: In total, 702 primary care nurses participated in the study. Responses were obtained from 14 out of the 17 Spanish Autonomous Communities. Nursing foundation for quality of care, management and leadership of head nurse and nurse–physician relationship were identified as strengths, whereas nurse participation in center affairs and adequate human resources to ensure quality of care were identified as weaknesses of the nursing work environment in primary health care. Older nurses and those educated to doctoral level were the most critical in the nursing work environments. Variables Age, Level of Education and Managerial Role showed a significant relation with global score in the questionnaire. Conclusion: Interventions by nurse managers in primary health care should focus on improving identified weaknesses to improve quality of care and health outcomes.

## 1. Introduction

The nursing workforce plays a crucial role in health systems globally. According to the World Health Organization (WHO) [1], nurses must work to their full potential if countries are to achieve universal health coverage for the population. Nurses are a key element in the sustainability of the health service, enhancing quality of care and promoting patient safety, satisfaction and confidence [2,3]. These reasons have justified the implementation of the international campaign “Nursing Now” and the declaration of the year 2020 as the International Year of the Nurse.

In order to achieve the highest possible quality of nursing care, it is essential to analyze the elements contributing to the delivery of such care; the sum of these elements is the nursing work environment (NWE). Thus, NWE is defined as the characteristics of the workplace that promote or hinder the delivery of quality nursing care [4]. Positive NWEs are characterized by lower rates of mortality, morbidity and adverse events [5,6,7]; reduced administrative costs and absenteeism [8]; a lower level of burnout and increased patient satisfaction with nursing care and the health organization [9]. Improving health outcomes, the sustainability of the health system and user satisfaction are common goals for healthcare managers, which can be addressed from the perspective of NWE with positive conditions for care. The Magnet Hospital program, for example, accredits centers of excellence and quality of care based on evidence-based practice [10,11,12]. Magnet centers promote the development of a culture of safety and quality care, staff recognition programs, interdisciplinary communication and horizontal management that contribute positively to the work environment [13,14].

NWE studies have been extensive in hospital settings [15]. However, there is a lack of evidence about the impact of positive primary health care (PHC) NWE on patient outcomes. This may be a handicap for optimizing PHC services, where the configuration of the microenvironment should be conducive to the provision of excellent care. Previous studies [4] have pointed to a possible association between positive PHC NWE and patient outcomes, but the evidence is still scarce.

Previous studies [4,16,17,18] have assessed the PHC NWE in Spain, concluding that management and leadership of the head nurse, nurse–physician relationship and nursing foundation for quality of care are the most highly valued aspects by primary care nurses, whereas adequate human resources to ensure quality of care is frequently among the least valued characteristics of the workplace. A recent scoping review of the literature suggested that the reality of the PHC NWE is different from that of the hospital NWE. This is due to differences in the decision-making and organizational processes and the relationships between team members. Thus, the evidence available on the hospital NWE may not be applicable to the reality of the PHC NWE [19]. According to Lucas and Nunes [19], the work environment is the most influential factor with the greatest impact on nursing outcomes and on the perceptions of the quality of care and client safety. In order to assess and thus improve the quality of the NWE, Poghosyan et al. [20] developed a global model for optimizing the PHC NWE. In their model, they propose the integration of institutional policies, organizational innovation and research.

We argue that positive PHC NWE can increase the quality of nursing care and, subsequently, improve patient outcomes. However, in order to achieve this goal, it is necessary to analyze the strengths and weaknesses of the PHC NWE. Therefore, we aim to analyze the characteristics of NWE in PHC settings in Spain, identifying these environments’ strengths and weaknesses. In addition, we aim to analyze the associations between sociodemographic and professional variables in our sample, as well as the nursing professionals’ perception of their NWE.

## 2. Materials and Methods

### 2.1. Design

Cross-sectional study of the strengths and weaknesses of NWE in PHC settings in Spain.

### 2.2. Participants and Study Location

We recruited a non-probabilistic, multi-stage sample of qualified nurses working in PHC settings in Spain. Due to the limited resources for data collection and the geographical dispersion of PHC professionals, the maximum possible number of participants was proposed as a sampling target during the set data collection period. Inclusion criteria for participation in this study were: being qualified as a nurse, having worked in the same PHC setting for at least 3 months and signing the consent form. Data collection was paused during vacation periods in order to limit the possibility of bias arising from staff turnover.

A total of 817 questionnaires were received, of which 115 (14.7%) were excluded. The reasons for their exclusion were: 7 (6.09%) nurses who did not work at PHC, 27 (23.48%) did not complete the questionnaires correctly and 81 (70.43%) did not meet the criteria for participation in the study.

First, the researchers disseminated the study through social networks (Twitter, Facebook) obtaining an unrestricted sample [21]. Simultaneously, direct dissemination among PHC nurses was achieved by contacting key informants via institutional e-mail (on two occasions, one month apart) to encourage and remind them to participate in the study. This was done between June 2018 and June 2019.

A data collection pack containing the consent form, a questionnaire of sociodemographic variables developed ad hoc and the Practice Environment Scale of the Nursing Work Index (PES-NWI) questionnaire was sent via email to the participants. We used Google Forms^®^, with a limited response via Internet Protocol (IP), to collect the data electronically. A letter of invitation to participate in the study was also attached, as well as a request to contribute to the dissemination of the study with the purpose of reaching a greater number of PHC nurses through a snowball sampling technique.

### 2.3. Instruments of Data Collection

An ad hoc questionnaire was designed to collect sociodemographic and professional variables including age, gender, level of education, professional specialization, work experience in PHC settings and management role and responsibilities.

There are multiple measurement tools for the study of NWE [22]. The PES-NWI is among the most widely used and has high levels of consistency and reliability (Cronbach’s Alpha 0.807–0.916) [9,23,24]. We used the Spanish version of the PES-NWI, which was adapted and validated for use in Spanish PHC settings by de Pedro-Gómez et al. [25] in 2012. The Spanish PES-NWI is divided into 5 dimensions, namely, nurse participation in center affairs (nine items) (D1), nursing foundation for quality of care (10 items) (D2), management and leadership of head nurse (five items) (D3), adequate human resources to ensure quality of care (four items) (D4) and nurse–physician relationship (three items) (D5). This tool includes a total of 31 items measured on a 4-point Likert scale, with scores ranging between 4 to 124. A favorable environment receives scores of >2.6, neutral or controversial environments receive scores between 2.6 and 2.4 and an unfavorable NWE receives scores of >2.4 for each dimension. Total scores are interpreted as follows: values ≥80.6 are interpreted as positive environments for nursing work, values between 74.5 and 80.5 are identified as controversial environments and values ≤74.4 are classed as negative environments for nursing work [26]. For the present study, Cronbach’s alpha for the PES-NWI was 0.937, with a reliability range between 0.836 and 0.935 for each dimension.

More recently, an abbreviated version of the PES-NWI tool was developed by Gea-Caballero et al. [26] with the aim of synthetizing and prioritizing the essential elements for improving PHC settings in the Spanish context. The TOP10 Questionnaire of Assessment of Environments in Primary Health Care (hereinafter TOP10) is divided into 3 dimensions, namely, nurse participation in center affairs (D1a), quality of care (D2a) and human resources (D3a). It comprises 10 items identified as the “essential elements of care”; if not positive, these essential elements of care can seriously affect the quality of care in any given NWE. The total score of the TOP10 questionnaire ranges between 10 and 40. The higher the score, the more favorable the NWE. Cronbach’s alpha for the TOP10 questionnaire was reported at 0.816 in a previous study [26]; for the present study, Cronbach’s alpha was 0.805. The TOP10 questionnaire was completed by the researchers based on the results obtained from the Spanish version of the PES-NWI tool.

The information for the analysis of the TOP10 was extracted from the PES-NWI questionnaire, since the 10 essential items are part of the 31 that make up the PES-NWI.

### 2.4. Statistical Analysis

We used descriptive statistics to analyze the sociodemographic and professional characteristics of our sample. Mean and standard deviation for quantitative variables and frequencies, and percentages for qualitative variables, were calculated.

We calculated the Cronbach’s alpha coefficient of both the PES-NWI and TOP10 questionnaire to assess internal consistency and reliability as a whole, and for each dimension separately. Reliability was considered excellent when Cronbach Alpha was greater than 0.90; good between 0.80 and 0.89; acceptable between 0.70 and 0.79; questionable between 0.60 and 0.69; poor between 0.50 and 0.59 and unacceptable below 0.50 [27].

We carried out a normality test using Shapiro–Wilk and Kolmogorov–Smirnov tests as appropriate for the quantitative variables. Four variables were found to have a non-normal distribution, namely, dimensions D1, D3, D4 and D5, and were analyzed using non-parametric methods. For the bivariate analyses of variables with a normal distribution (D2 and TOP10), we used a *t*-test for independent samples for dichotomous variables and ANOVA for polytomous variables. If significant differences in the ANOVA were found, the Bonferroni test was applied to determine which pairs of categories presented significant differences between them.

The SPSS v23 statistical package was used for the statistical analysis, and a significance level of *p* = 0.05 was adopted.

### 2.5. Ethical Considerations

We safeguarded our participants’ confidentiality and anonymity according to Spanish/European data protection regulations (Organic Law 3/2018). The participants were informed about the methods and aims of the study and gave their consent to take part in this investigation. This study was approved by the Research Ethics Committee of the Valencian Community (Xàtiva/Ontinyent, Valencia, Spain). The participants did not receive any compensation for completing the questionnaires.

## 3. Results

The final sample consisted of 702 qualified PHC nurses. The participants were mainly women (71.9%), aged 40 years or older (61.1%) and with more than 10 years of work experience (52.4%). Most of our participants were educated to degree level (64.3%), 5.4% were nurse specialists and 11.8% were nurse managers or coordinators. Responses were obtained from 14 out of the 17 Spanish Autonomous Communities, with a greater representation from the Valencian Community and the Canary Islands (76.3%).

The results of the PES-NWI are shown in Table 1. The NWE in Spanish PHC settings was positive with a total average score of 82.4. Three dimensions were identified as strengths, namely, nursing foundation for quality of care (D2), management and leadership of head nurse (D3) and nurse–physician relationship (D5), and one dimension was identified as a weakness in the PHC settings studied: adequate human resources to ensure quality of care (D4). Nurse participation in center affairs (D1) was considered as neutral or controversial. Specifically, 17 items were identified as strengths, and only 9 were classed as weaknesses of the PHC settings (items 6, 7, 9, 12, 16, 25–28). The reliability of the PSE-NWI in our sample was confirmed with a Cronbach’s alpha 0.937 and a range between 0.836 and 0.935 for each of its dimensions.

The average score from the TOP10 questionnaire was 29.7. Two of its dimensions were identified as strengths: participation in center affairs (D1a) and quality of care (D2a), and one was identified as a weakness: human resources (D3a). The reliability of the TOP10 tool in our sample was confirmed with a Cronbach’s alpha 0.805.

The bivariate analysis of the results from PES-NWI and the sociodemographic and professional variables are shown in Table 2. Age was statistically significant for the PES-NWI and 3 out of its 5 dimensions (D2, D3 and D5). The participants’ level of education was statistically significant for the overall PES-NWI and 4 of its dimensions (D1, D2, D3 and D5), and management role was found to be statistically significant for the PES-NWI and 4 of its dimensions (D1, D2, D3 and D4). We did not find a statistically significant correlation between PES-NWI and the rest of the sociodemographic and professional variables.

We investigated gender inequalities in our sample (Table 3). Our results show that most of the nurses were aged ≤50 years (*p* = 0.031). The representation of women at higher education levels (masters or doctorate) is proportionally lower than that of men (*p* = 0.024) but, paradoxically, women achieve a higher percentage of specialist training (*p* = 0.048).

## 4. Discussion

The main objective of this study was to identify strengths and weaknesses in the PHC work environment in Spain. Our results suggest that positive NWEs in PHC in Spain are characterized by nursing foundations for quality care (D2), management and leadership of the head nurse (D3) and the nurse–physician relationship (D5). This is consistent with previous studies in our context [16,17]. Nurse participation in center affairs (D1) was identified as neutral or controversial in our study as opposed to a previous study by Gea-Caballero et al. [16], where it was identified as a strength. Overall, our results differ from those obtained by de Pedro-Gómez et al. [28], who classified the NWE in PHC settings in the Balearic Islands as controversial (80.4 points).

This study fits well with the improvement model proposed by Poghosyan et al. [20]. Political decision-making and organizational innovation in PHC settings are key to improve identified weaknesses. Furthermore, research in healthcare settings is essential to not only increase knowledge of, and improve, both processes and procedures, but also to create an organizational culture that promotes the integration of the best available evidence [18], thus improving patient outcomes and increasing service user satisfaction.

Given the difficulty in finding other studies in the PHC setting, and due to their conceptual proximity, we compared our results with those reported in studies about magnet hospitals (as described in the introduction). Our results show that dimensions D2, D3 and D5 are associated with a positive NWE. This is in agreement with the results from previous studies carried out in “non-magnet hospitals”. These are encouraging findings, but they are still far from those obtained in “magnet hospitals”, where every single dimension of the PES-NWI was identified in historical studies as a strength [29,30]. This is an encouraging finding as it demonstrates that the transformation of weak or controversial dimensions into strengths is possible, as evidenced by the results obtained in magnet and excellent work environments.

In Spanish hospitals, the same three dimensions, namely, D2, D3 and D5, were shown to be neutral or controversial for the NWE [31]. This diversity in the results suggests that the quality of the NWE in the hospital context depends on external as well as internal characteristics of the healthcare service. Therefore, interventions to improve the NWE in the PHC context should be individualized and based on the results obtained from each separate healthcare institution (microenvironment). The same reflection is applicable to PHC work environments. However, a study by de Pedro et al. [28] identified D2 and D3 only as strengths in hospitals with 300–500 beds. Paradoxically, in international studies about the characteristics of the NWE, we find a greater diversity of scenarios; some identify all the dimensions of the PES-NWI as strengths [32,33], others show management and leadership of the head nurse and nurse–physician relationship as strengths (D3 and D5) [34], and some identify management and leadership of the head nurse as the only strength (D3) [35].

In the PHC NWE in Spain, the size of the workforce or human resources (D4) is identified as a clear weakness, coinciding with national studies in both PHC [16,17,28] and hospital [28,31,36] work environments. These results coincide with those portrayed in international studies about NWE in the hospital setting [34,35]. The comparative studies between “magnet and non-magnet hospitals” reported similar results in historical studies, with human resources (D4) being identified as a weakness in “non-magnet hospitals”; it was not identified as a weakness in “magnet hospitals”, but it was the worst valued dimension [29,30]. This same situation was also observed in international studies, both European [32] and Asian [33], with human resources usually being the worst valued dimension. The problem with human resources is particularly serious in Spain, where the nurse–patient ratio is 567 per 100,000 inhabitants, well below the European average (811/100,000) and far from the more industrialized countries, such as Finland, Denmark or Belgium (1500/100,000) [37]. Despite the efforts made in recent years to increase the nursing workforce, and the commitment to nurses as health agents, it is still a limitation that compromises patient safety and quality of care. Furthermore, the nurse–physician ratio in Spain is severely unbalanced. According to the Organisation for Economic Co-operation and Development (OCED) [38], the number of physicians per inhabitant in Spain is above average (7th place and above countries such as Italy, Australia, France and Finland), but the number of nurses is well below the average worldwide (23rd place out of 26 countries). Finland and Germany triple the number of nurses in Spain, and Norway quadruples it. It should not be forgotten that there is a direct correlation between the ratio of nurses and patient mortality, as well as other unwanted events and health outcomes [5,6,7,8,9].

These facts, framed in a global SARS-COV-2 pandemic, reveal and exacerbate existing problems within the healthcare service. For example, the COVID-19 pandemic has added undue pressure to the health services in Spain, thus highlighting the lack of qualified nurses. As suggested by Seccia Ruggero [39], the replenishment of material resources can be achieved relatively easily, but reinforcement with qualified nurses is difficult to achieve and cannot be done over a short period of time. An adequate nursing provision could contribute to improved outcomes in health crises, such as at the peak of the COVID pandemic, which has led the WHO in April 2020 to call for more investment in nurses [40]. Key stakeholders and those responsible for decision making on healthcare service planning should consider the need to increase the Spanish nursing workforce and draw a plan accordingly in the years to come.

The results from the TOP10 scale [26] were fully consistent with the PES-NWI results, identifying participation of nurses in the affairs of the center (D1a) and the nursing foundations for quality of care (D2a) as strengths, and human resources (D3a) as a weakness. We argue that TOP10 is a simpler way of identifying the strengths and weaknesses associated with the NWE, making it easier and simpler for nurse managers to identify weakness or areas for improvement within their PHC work environments. In addition, the results from the TOP10 scale are valid and reliable as supported by a recent study by Martínez-Riera et al. [41], where a group of community care experts considered that 9 out of the 10 items of the TOP10 scale were essential elements to the PHC NWE.

The comparative study of the sociodemographic and professional variables and the perception of the PHC NWE shows significant differences associated with age, level of education and the level of management in which the professionals were involved. Older nurses (50+) were the most critical with their work environments. In addition, significant differences were found for dimensions D2, D3 and D5 separately. The age-related differences found in Spanish studies should be assessed with caution due to the average age difference between nurses employed in public and private services, with greater representation in privately managed centers of the age range under 40 years [16].

Nurses educated to doctoral level identified the Spanish PHC NWE as a negative environment for nursing care and also pointed to the dimensions of nursing participation in center affairs (D1), nursing foundations for quality of care (D2) and human resources (D4) as weaknesses. Interestingly, the dimension nursing foundations for quality of care (D2) was identified as a weakness by doctoral nurses and as a strength by the rest of the nursing professionals. The same was observed in a previous study [17] in the Community of Madrid. This may suggest a lower level of job satisfaction among the most the nursing professionals with a highest level of education [33], or perhaps it may reflect a greater capacity for critical thinking. This situation is paradoxical. PHC nurses look after an ageing population with highly complex and chronic conditions. Thus, it would seem reasonable to integrate nurses with high levels of training and those in advanced practice roles in PHC settings [42]. The International Council of Nurses [43] defines advanced nurse practitioners as professionals who have acquired the theoretical knowledge, complex decision-making skills and clinical competencies for extended practice in the country and context for which they are accredited. Advanced training, such as a master’s or doctoral degree, is recommended for an advanced nursing practice qualification [44]. Our results show that highly qualified nurses (doctoral level) value their work environment the least, reflecting the fact that the work environment may not be adapted to the academic level of these professionals. We argue that it is necessary to ensure that highly qualified nurses and those in advanced practice roles [45] are able to work to their full potential within PHC settings in Spain, and recommend that aspects such as the nurses’ level of training and expertise, and not simply their seniority and years of experience, are taken into account when designing nursing career pathways. Advanced practice nursing.

The nurses in a management role identified nursing participation in the affairs of the center (D1), nursing foundations in the quality of care (D2), management and leadership of the head nurse (D3) and human resources (D4) as strengths, with their score being higher than that of their staff nurse colleagues. This was also the case in previous studies carried out in Spain [16,17]. Interestingly, the human resources dimension (D4), which was recognized as a weakness in our study, as well as in previous studies [16], was not identified as such by the nurse managers, who considered it to be a strength. We believe that this phenomenon should be analyzed further through in-depth qualitative interviews with nurse managers, as well as other key stakeholders, in order to fully understand the root cause of this problem. Namely, it is possible that there are specific factors which are affecting the participants’ assessment of the impact of the nursing workforce on the NWE. This may include the level of participation of the highest trained professionals and the quality of the relationship between the nurse managers and the rest of the staff.

Finally, from a gender perspective, no significant differences were observed when comparing the NWE with the gender of the participants in our study.

### Limitations

We wish to highlight a number of limitations. First, our cross-sectional design does not allow us to infer causality in the relationships between variables. Second, although our sample is larger than that of previous PHC NWE studies, we cannot guarantee the representativeness of the entire nursing population in Spain as some of the Spanish territories are either not represented or under-represented. Third, although precautions were taken to control for duplicate responses, it is possible that some scaped our scrutiny. For these reasons, we recommend that further studies analyzing the NWE in PHC settings with more powerful samples are carried out in order to confirm these data.

## 5. Conclusions

The NWE in PHC settings in Spain is positive and comparatively better than the NWE in hospital settings. We identified the following strengths: (1) nursing foundation for the delivery of care, (2) management and leadership of the head nurse and (3) nurse–physician relationship, and the following weakness: (1) participation of nurses in the affairs of the center and (2) human resources. We argue that there is room for improvement of the NWE in PHC settings in Spain, and that efforts should be directed towards the neutral and negative aspects identified. Two groups of nurses were particularly critical of their NWE, namely, older nurses and those educated to doctoral level. Nurse managers did not identify human resources as a weakness, contrary to the results from previous national and international investigations. We found no evidence of gender influence on the results obtained.

## Figures and Tables

**Table 1 ijerph-18-00434-t001:** Total scores by dimensions and items from the PES-NWI.

	Score Mean (SD)		
	Strengths	Neutral	Weaknesses
Dimension 1: Nurse participation in the center affairs		2.50 (0.7)	
1 *: Staff nurses are formally involved in the internal management of the center (boards, decision-making bodies)		2.47 (0.9)	
2: Nurses at the center have opportunities to participate in decisions affecting the various policies developed by the center		2.47 (0.9)	
3: Many opportunities exist for the professional development of nurses		2.41 (0.9)	
4: Management listens and responds to the concerns of its nurses		2.55 (0.9)	
5: The Director of Nursing is accessible and easily “visible”	2.90 (1.0)		
6: A professional career can be developed or there are opportunities for promotion in the clinical career			2.27 (1.0)
7: Managers consult with nurses about problems and ways of doing things on a day-to-day basis			2.34 (1.0)
8: Staff nurses have opportunities to participate in the center’s committees, such as the committee on research, ethics, infections	2.87 (0.9)		
9: Nursing managers are at the same level of power and authority as other managers in the center			2.34 (1.0)
Dimension 2: Nursing foundation for quality of care	2.72 (0.6)		
10: Nursing diagnostics are used	2.88 (1.0)		
11 *: There is an active quality assurance and improvement programme		2.59 (0.9)	
12: There is a programme for welcoming and mentoring new nurses			2.25 (1.1)
13: Nursing care is based on a nursing model rather than a biomedical model	2.68 (0.9)		
14 *: Assigning patients to each nurse promotes continuity of care	3.14 (0.9)		
15 *: There is a common, well-defined nursing philosophy that permeates the environment in which patients are cared for	2.63 (0.9)		
16: There is a written and updated plan of care for each patient		2.44 (0.9)	
17: Center managers are concerned that nurses provide high-quality care	2.58 (0.9)		
18 *: A program of continuing education is developed for nurses	2.97 (0.9)		
19 *: The nurses in the center have adequate clinical competence	3.03 (0.8)		
Dimension 3: Management and leadership of head nurse	2.93 (0.9)		
20*: The coordinator/supervisor is a good manager and leader	2.88 (1.0)		
21: The supervisor/coordinator supports the staff in their decisions, even if the conflict is with medical staff	2.94 (1.0)		
22: The supervisor/coordinator uses mistakes as opportunities for learning and improvement, not as criticism	2.87 (1.0)		
23: The supervisor/coordinator is sympathetic and advises and supports the nurses	3.07 (1.0)		
24: Work well done is recognised and praised	2.90 (1.0)		
Dimension 4: Adequate human resources to ensure quality of care			2.33 (0.8)
25 *: There are enough employees to do the job properly			2.28 (1.0)
26 *: There are sufficient numbers of registered nurses to provide quality care			2.35 (1.0)
27: Support services (wardens, administrative staff, etc.) are adequate and make it easier to spend more time with patients			2.38 (0.9)
28: There is sufficient time and opportunity to discuss care issues with the other nurses			2.32 (0.9)
Dimension 5: Nurse–physician relationship	2.87 (0.7)		
29: A lot of teamwork is done between doctors and nurses	2.63 (0.9)		
30: There are good working relationships between doctors and nurses	3.10 (0.7)		
31 *: Practice between nurses and doctors is based on appropriate collaboration	2.87 (0.8)		
**TOP10 questionnaire**	Strengths	Neutral	Weaknesses
Dimension 1a: Participation in center affairs	2.73 (0.7)		
Dimension 2a: Quality of care	2.92 (0.7)		
Dimension 3a: Human resources			2.31 (1.0)
**Overall result**			
PES-NWI total	82.43 (17.4)		
TOP10 total	29.68 (6.2)		

SD: standard deviation; strength: score > 2.6; neutral or controversial: score between 2.4–2.6; weakness: score < 2.4; positive environment: PES-NWI score > 80.6; controversial environment: PES-NWI score 74.4–80.6; negative environment: PES-NWI score < 74.4; * TOP10 items.

**Table 2 ijerph-18-00434-t002:** Bivariate analysis of the results from PES-NWI and the sociodemographic and professional variables.

		Scoring M(SD)					
		Global	Dimensions				
			D1	D2	D3	D4	D5
Age (years)	*p* ^a^	0.016 *	0.123	0.047 *	0.041 *	0.467	0.000 *
Less than or equal to 30		84.98 (15.9)	2.63 (0.6)	2.74 (0.6)	3.00 (0.8)	2.44 (0.7)	3.04 (0.7)
31–40		84.99 (17.7)	2.55 (0.6)	2.83 (0.7)	3.06 (0.9)	2.31 (0.8)	3.06 (0.7)
41–50		81.56 (17.3)	2.48 (0.7)	2.70 (0.6)	2.92 (0.9)	2.29 (0.8)	2.82 (0.7)
>50		80.22 (17.6)	2.46 (0.7)	2.65 (0.6)	2.82 (0.9)	2.33 (0.8)	2.71 (0.7)
Managerial role	*p* ^b^	<0.000 *	<0.000 *	<0.000 *	<0.000 *	<0.000 *	0.296
Yes		94.65 (14.4)	3.01 (0.6)	3.06 (0.5)	3.52 (0.6)	2.63 (0.8)	2.96 (0.7)
No		80.55 (17.3)	2.44 (0.6)	2.67 (0.6)	283 (0.9)	2.28 (0.8)	2.86 (0.7)
Level of education	*p* ^a^	<0.000 *	0.026 *	<0.000 *	0.011 *	0.518	0.017 *
Diploma		84.73 (17.4)	2.56 (0.7)	2.82 (0.6)	3.02 (0.9)	2.36 (0.8)	2.95 (0.8)
Degree		81.21 (16.1)	2.52 (0.6)	2.67 (0.6)	2.89 (0.9)	2.28 (0.7)	2.76 (0.7)
Specialisation		80.91 (16.1)	2.53 (0.6)	2.58 (0.6)	2.93 (0.8)	2.39 (0.7)	2.73 (0.7)
Master		80.49 (17.7)	2.45 (0.6)	2.66 (0.6)	2.81 (0.9)	2.30 (0.8)	2.83 (0.7)
PhD		71.13 (16.5)	2.16 (0.7)	2.27 (0.6)	2.53 (0.8)	2.09 (0.7)	2.63 (0.7)

M: mean; SD: standard deviation; D1: participation of nursing staff; D2: nursing foundation in quality of care; D3: capacity, leadership and support of nursing staff by managers; D4: the size of staff and adequacy of human resources; D5: relationships between nursing and medical professionals; * significant values *p* < 0.05; *p*
^a^: ANOVA; *p*
^b^: Student’s *t*-test.

**Table 3 ijerph-18-00434-t003:** Distribution and comparison of socio-demographic and occupational data by gender.

		Gender	
		Man (%)	Woman (%)
Age (years)	*p* = 0.031 *		
less than or equal to 30		22.7	77.3
31–40		28.6	71.4
41–50		22.3	77.7
>50		34.0	66.0
Level of education	*p* = 0.024 *		
Diploma		29.6	70.4
Degree		21.8	78.2
Specialisation		16.5	83.5
Master		29.3	70.7
PhD		46.7	53.3
Specialist training	*p* = 0.048 *		
Yes		15.8	84.2
No		20.7	79.3
Working experience (years)	*p* = 0.206		
<2		22.0	78.0
2–4		21.6	78.4
5–10		31.2	68.8
>10		29.3	70.7
Managerial role	*p* = 0.204		
Yes		21.7	28.4
No		78.3	71.6

*p*: chi square test; * *p* < 0.05

## Data Availability

Not applicable.

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
