# Peer review of "Study of the Strengths and Weaknesses of Nursing Work Environments in Primary Care in Spain"

_ijerph, 2021, doi:10.3390/ijerph18020434_

Round 1

Reviewer 1 Report

Abstract: Add a brief description of the sample.

Methodology:
Describe the recruitment of the sample in more detail, and justify why you used the "a non-probabilistic, multi-stage, convenience sample" methods in your quantitative study. Why the sample of primary nurses was assembled by this method and also first through social networks - although primary nurses in Spain are available for random or absolute selection, because they are working and are organized in Healthcare Centers under the National Health System. Like the snowball sampling technique, these methods are suitable for qualitative rather than quantitative research.

The authors should also indicate the total number of primary nurses in Spain (0.65 per 1000 inhabitants?). And also from which parts of Spain the nurses were recruited for this study (76.3% being the Valencian Community and the Canary Islands?). In terms of population, these areas make up only 15% of Spain.

The 1. Introduction should also describe the working environment of primary nurses in Spanish Healthcare Centers.

Discussion:
Why do the authors mention in discussion the SARS-COV-2 pandemic, when data for this cross-sectional study were collected from 2018 to 2019?

Author Response

REV1

We appreciate your valuable comments, which we are sure have improved the final version of the manuscript.

Abstract: Add a brief description of the sample.

We have added the information as requested.

Methodology:
Describe the recruitment of the sample in more detail, and justify why you used the "a non-probabilistic, multi-stage, convenience sample" methods in your quantitative study. Why the sample of primary nurses was assembled by this method and also first through social networks - although primary nurses in Spain are available for random or absolute selection, because they are working and are organized in Healthcare Centers under the National Health System. Like the snowball sampling technique, these methods are suitable for qualitative rather than quantitative research.

We confirm that our sampling strategy was non-probabilistic, multi-stage (the word convenience is redundant and has been deleted from the manuscript) (line 86). Recruitment took place at different moments throughout a period of over a year. Initially, we disseminated our study through social media, but this strategy failed to recruit a sufficient number of participants. Our second recruitment strategy involved using the mailing list of a national scientific society, which gave its permission to disseminate the study aims and procedures. The third stage in involved the participation of key stakeholders (community nurses from a range of autonomous communities in Spain) in the process of recruitment through a snowballing technique, who kindly forwarded information about the study to their contacts.

The authors should also indicate the total number of primary nurses in Spain (0.65 per 1000 inhabitants?). And also from which parts of Spain the nurses were recruited for this study (76.3% being the Valencian Community and the Canary Islands?). In terms of population, these areas make up only 15% of Spain.

Unfortunately, we have not been able to find any official source or study that estimates the ratio of nurses per patient in primary care settings in Spain.

This study was design by researchers from the Autonomous Community of Valencia, which is the reason why this area is so well represented. The snowballing technique in the Canary Islands was very successful, which explains why this geographical area is also well represented. In the rest of the autonomous communities the rate of response was more discreet, despite the fact that the recruitment period was long; this explains why some areas are underrepresented in our sample. We know that this limits the representativeness and generalizability of our results and, for this, reason, we have highlighted it in the limitations section.

The 1. Introduction should also describe the working environment of primary nurses in Spanish Healthcare Centers.

Many thanks for this comment. We have added a new paragraph in the introduction (lines 63-75) describing the Spanish primary care nursing work environments (although research in this area in rather scarce).

Discussion:
Why do the authors mention in discussion the SARS-COV-2 pandemic, when data for this cross-sectional study were collected from 2018 to 2019?

Indeed, we collected our data from 2018 to 2019, well before the outbreak of the pandemic. Our reason for referring to the current situation is to highlight the fact that the existing problems and shortcomings of the primary care nursing work environment, already identified by our participants, have been exacerbated by the pressures added by COVID19. Further, the current situation of the primary care services in Spain has done nothing but confirmed the nurses’ perception of the weaknesses of their work environment and, for the first time, the nurse-patient ratio in our country is in the spotlight and currently being discussed at a high level by our congressmen and women. This is the reason why we have chosen to add this comment to the discussion; we feel that this is no longer a perception nor a corporatist response of primary care nurses, but a real social and health problem (lines 253-254)

Reviewer 2 Report

Thank you for the opportunity to review this study of strengths and weaknesses of nursing work environments in primary care in Spain. Overall I found the paper interesting and results and conclusions sound. There are a couple of points which might benefit from increased clarity.

The abstract could contain more detail on analyses.

By duplicate answers do you mean people submitted the questionnaire twice?

Quite a high proportion of questionnaires did not met criteria for participants, was this due to using social media Twitter to  recruit

Can you provide a little more detail on how the researchers completed the TOP10 questionnaire (page 3 line124)

The second paragraph on page 7 line 204 ‘In Spanish hospitals…’ might benefit from rewriting to place the implications for primary care first.

There are a few spelling typographical mistakes

Abstract ‘Stregnths’

  page 4 line167 ‘dimentions’

Author Response

Thank you for the opportunity to review this study of strengths and weaknesses of nursing work environments in primary care in Spain. Overall I found the paper interesting and results and conclusions sound. There are a couple of points which might benefit from increased clarity.

Thank you, we appreciate your valuable comments. We have proceeded to incorporate your suggestions and correct the errors detected. We are convinced that the suggestions have contributed to improving the manuscript.

The abstract could contain more detail on analyses.

Thanks for the comment, the description of the sample and the relations of the variables with the total score of the questionnaire have been added in the abstract.

By duplicate answers do you mean people submitted the questionnaire twice?

Thanks for the important appreciation, it is a mistake and has been corrected, including the correct information: nurses who did not work at PHC

Quite a high proportion of questionnaires did not met criteria for participants, was this due to using social media Twitter to recruit

Thank you. It may be for what you suggest, but not necessarily for that. Most of the questionnaires that did not meet the participants' criteria were the result of having short temporary contracts in the company, not reaching the minimum experience to be included.

Can you provide a little more detail on how the researchers completed the TOP10 questionnaire (page 3 line124)

Since the TOP10 questionnaire was constructed by extracting 10 items from the 31 that make up the PES-NWI, the participants were administered the complete 31-item questionnaire and for the analysis the calculations were made for the entire questionnaire, and on the other hand only for the 10 items qualified as essential (TOP10).

We added:

The information for the analysis of the TOP10 was extracted from the PES-NWI questionnaire, since the 10 essential items are part of the 31 that make up the PES-NWI.

The second paragraph on page 7 line 204 ‘In Spanish hospitals…’ might benefit from rewriting to place the implications for primary care first.

Thank you, the text has been reordered following your suggestion, and is as follows:
In Spanish hospitals, the same three dimensions, namely D2, D3 and D5, were shown to be neutral or controversial for the NWE [31]. This diversity in the results suggests that the quality of the NWE in the hospital context depends on external as well as internal characteristics of the healthcare service. Therefore, interventions to improve the NWE in PHC context should be individualisaed and based on the results obtained from each separate healthcare institution (micro-environment). The same reflection is applicable to primary care work environments. However, a study by de Pedro et al [28] identified D2 and D3 only as strengths in hospitals with 300-500 beds. Paradoxically, in international studies about the characteristics of the NWE we find a greater diversity of scenarios; some identify all the dimensions of the PES-NWI as strengths [32,33], others show management and leadership of the head nurse and nurse-physician relationship as strengths (D3 and D5) [34], and some identify management and leadership of the head nurse as the only strength (D3) [35].

There are a few spelling typographical mistakes

Thank you, a complete external language review has been conducted.

Abstract ‘Stregnths’

Thank you, the error has been modified.

page 4 line167 ‘dimentions’

Thank you, the error has been modified.

Round 2

Reviewer 1 Report

no comments